# POINT2PRIMITIVE: CAD RECONSTRUCTION FROM POINT CLOUD BY DIRECT PRIMITIVE PREDICTION

## ABSTRACT

Recovering CAD models from point clouds requires reconstructing their topology and sketch-based extrusion primitives. A dominant paradigm for representing sketches involves implicit neural representations such as Signed Distance Fields (SDFs). However, this indirect approach inherently struggles with precision, leading to unintended curved edges and models that are difficult to edit. In this paper, we propose Point2Primitive, a framework that learns to directly predict the explicit, parametric primitives of CAD models. Our method treats sketch reconstruction as a set prediction problem, employing a improved transformer-based decoder with explicit position queries to directly detect and predict the fundamental sketch curves (i.e., type and parameter) from the point cloud. Instead of approximating a continuous field, we formulate curve parameters as explicit position queries, which are optimized autoregressively to achieve high accuracy. The overall topology is rebuilt via extrusion segmentation. Extensive experiments demonstrate that this direct prediction paradigm significantly outperforms implicit methods in both primitive accuracy and overall geometric fidelity.

## 1 INTRODUCTION

Computer-Aided Design (CAD) reconstruction, which aims to convert unstructured data like point clouds into structured, parametric models, is a long-standing goal in computer graphics and engineering (Kruth and Kerstens, 1998). Among various modeling techniques, the sketch-and-extrude process is one of the most fundamental operations, capable of generating a vast array of mechanical parts and industrial designs (Willis et al., 2021). Therefore, automatically reversing this process, transforming a raw 3D scan into a structured sketch-based CAD representation, holds immense value for digital manufacturing, legacy part recreation, and design modification.

Despite recent progress driven by large-scale datasets (Willis et al., 2021; Koch et al., 2019), a critical challenge lies in accurately capturing the model's boundaries. The unstructured, noisy nature of point clouds contrasts sharply with the precise, topologically-defined structure of CAD models. A dominant paradigm in recent literature has been to represent 2D sketches using implicit neural representations such as Signed Distance Fields (SDFs) (Uy et al., 2022; Ren et al., 2022; Li et al., 2023a). While these methods are powerful for capturing complex, continuous surfaces, they remain limited in several critical aspects. First, SDFs are purely geometric fields that lack semantic structure, offering no direct correspondence to the primitive entities used in CAD. Second, as approximation-based representations, they inevitably introduce noise and blur fine geometric details, which makes it difficult to recover sharp corners or well-defined curves. Consequently, reconstructed sketches often exhibit **smoothed or rounded edges** rather than precise boundaries. Finally, because SDF-based sketches cannot be directly expressed in terms of editable primitives—**lines, circles, and arcs**, the lingua franca of CAD software—an additional conversion step is required. This transformation not only complicates the workflow but also accumulates parameter errors, further hindering faithful reconstruction and downstream editability.

The above limitations lead us to a fundamental question:

> *Can we bypass the imprecise, intermediate representations and **directly** predict the explicit, parametric sketch primitives from the point cloud?*

In this work, we affirmatively answer this question by proposing *Point2Primitive*, a framework grounded in a direct prediction methodology for CAD reconstruction from point clouds. Unlike indirect approaches that rely on intermediate surface fitting or voxelization, *our method directly targets the underlying parametric sketch primitives*. Specifically, we observe that many CAD models can be systematically deconstructed into a sequence of extrusion operations, each defined by a two-dimensional sketch—a closed profile formed by geometric primitives such as lines, arcs, and circles—together with an associated three-dimensional operation that specifies the extrusion direction and length.

Building on this insight, we reformulate the complex reverse engineering problem into two interdependent sub-tasks. First, in the stage of *topology reconstruction*, the input point cloud is partitioned into clusters that correspond to individual extrusions, a process we term *extrusion segmentation*. This step disentangles the overall geometry into meaningful sub-structures, enabling subsequent sketch prediction to operate on localized and semantically coherent point sets. Second, in the stage of *primitive prediction*, we recover the generating two-dimensional sketch for each segmented extrusion body. To achieve this, we design an improved transformer-based decoder that refines point features and directly decodes them into parametric primitives. The decoder integrates explicit geometric priors by formulating curve parameters as positional queries, which guide the attention mechanism and progressively update predictions across layers. This design enhances the accuracy and stability of primitive recovery.

Point2Primitive could bridge topology recovery and parametric sketch inference within a unified direct-prediction framework. By directly predicting the language of CAD, our method produces CAD models that are not only geometrically accurate but also truly editable. In summary, our main contributions are summarized as follows:

- We introduce Point2Primitive, a new pipeline that *directly* predicts parametric sketch primitives (i.e., lines, arcs, and circles) from raw point clouds, bypassing implicit fields and preserving sharp, semantically meaningful boundaries for downstream editability.

- We decompose reconstruction into (i) *extrusion segmentation* to recover topology by clustering the point cloud into extrusion bodies, and (ii) *primitive prediction* with an improved transformer decoder that uses explicit position queries and autoregressive parameter refinement to yield high-precision primitive types and parameters.

- Extensive experiments and ablations show consistent gains in boundary fidelity, primitive recovery, and editability over SDF-based and recent learning baselines, demonstrating both accuracy and robustness across diverse CAD geometries.

## 2 RELATED WORKS

### 2.1 CAD RECONSTRUCTION VIA PRIMITIVE DETECTION

Several learning-based approaches have also been proposed to fit geometric primitives to point clouds. These methods first segment the input points into clusters corresponding to the same surface primitives and fit the clustered points to the parametric surface primitives (Li et al., 2019; Sharma et al., 2020; Le et al., 2021; Li et al., 2023b) or free surfaces (Liu et al., 2023). Except for surface primitives, UCSG-Net (Kania et al., 2020) and CSG-stump (Ren et al., 2021) learn the CSG parse tree that interrelates the predicted geometry primitives to reconstruct shapes. The reconstructed CAD models are in B-rep format, which is not as easy to edit as the sketch-and-extrude modeling procedure. In this paper, we explore the inverse of the sketch-and-extrude process and strive to recover the CAD modeling procedure from the point cloud. Therefore, we segment the point clouds into clusters belonging to the same extrusion primitive and explicitly reconstruct every element of the extrusion primitives. Instead of fitting points to the parametric or free surface, we employ an improved transformer to learn and predict the curves directly.

### 2.2 SKETCH-AND-EXTRUDE CAD RECONSTRUCTION

Some new solutions use SDF to represent sketches following the recent implicit 3D shape representations (Park et al., 2019; Wang et al., 2022). ExtrudeNet (Ren et al.) and SECAD-Net (Li et al.,

2023a) can learn implicit sketches and differentiable extrusions. Point2Cyl reconstructs 3D shapes through sketch regression supervised by the latent embeddings of its SDF. However, these methods utilize the implicit fields for sketch representation, leading to curved edges of the reconstructed shapes. Meanwhile, the reconstructed sketch is hard to be directly edited because it is represented by the SDF. Further transformation from SDF to sketch further brings in extra parameter errors.

### 2.3 CAD GENERATION

Early attempts explored CAD sequence generation with VAE-based models (Wu et al., 2021; Kingma and Welling, 2022) and their extensions such as vector-quantized VAE (Xu et al., 2023) or multimodal diffusion frameworks (Ma et al., 2024). However, tokenizing CAD sequences often disrupts the underlying CSG structure (Yu et al., 2021), making it difficult for language models to capture topology. While SkexGen (Xu et al., 2022) addressed this by separately encoding topology and geometry, existing VAE–LM hybrids still generalize poorly to reconstruction tasks. Moreover, LLMs struggle with numerical precision (Yang et al., 2024), leading to parameter inaccuracies and limited transferability beyond training data.

## 3 METHOD

### 3.1 PRELIMINARIES

A 2D CAD sketch can be modeled as a finite sequence of geometric primitives $S_i = (c_1^{(i)}, \ldots, c_{N_i}^{(i)})$, where each primitive $c_j^{(i)} = (t_j^{(i)}, \rho_j^{(i)})$ comprises a type $t_j^{(i)} \in \mathcal{T}$ and a parameter vector $\rho_j^{(i)} \in \mathbb{R}^6$. We adopt three primitive types $\mathcal{T} = \{\mathsf{L}, \mathsf{A}, \mathsf{C}\}$ for *line*, *arc*, and *circle*, respectively. Given the point set of the $i$-th extrusion, $\mathbf{P}_{\mathcal{E}_i} \in \mathbb{R}^{N_{\mathcal{E}_i} \times 3}$, the network $f_\theta$ is trained to predict the sketch as a set/sequence of primitives:

$$f_\theta(\{\, \mathbf{p}_k \mid \mathbf{p}_k \in \mathbf{P}_{\mathcal{E}_i} \,\}) \;\approx\; S_i \;=\; (\, c_1^{(i)}, \ldots, c_{N_i}^{(i)} \,), \quad c_j^{(i)} = (t_j^{(i)}, \rho_j^{(i)}),\; t_j^{(i)} \in \{\mathsf{L}, \mathsf{A}, \mathsf{C}\}. \quad (1)$$

**Center-prior parameterization.** We use a center-prior parameterization with a fixed 6-D vector $\rho = (\rho_1, \ldots, \rho_6)$ and pad unused entries by $-1$. *Line* ($\mathsf{L}$): midpoint $\mathbf{m} = (x_m, y_m)$ and one endpoint $\mathbf{d}_1 = (x_1, y_1)$, encoded as $\rho = (x_m, y_m, x_1, y_1, -1, -1)$. *Arc* ($\mathsf{A}$): midpoint $\mathbf{m} = (x_m, y_m)$, start point $\mathbf{s} = (x_1, y_1)$, and end point $\mathbf{d}_2 = (x_2, y_2)$, encoded as $\rho = (x_m, y_m, x_1, y_1, x_2, y_2)$. *Circle* ($\mathsf{C}$): center $\mathbf{c} = (x_c, y_c)$ and radius $r$, encoded as $\rho = (x_c, y_c, r, -1, -1, -1)$. Unless otherwise stated, we refer to a primitive by $c = (t, \rho)$. The first two entries of $\rho$ (i.e., the center or midpoint) are further used as positional embeddings to guide attention in the improved transformer decoder (see Sec. 3.4).

**Sketch–extrude–shape relation.** An extrusion is denoted by $\mathcal{E}_i = [S_i, E_i]$, where $E_i$ is the associated operation. A solid shape is a finite collection of extrusions, $\mathbb{S} = \{\mathcal{E}_1, \ldots, \mathcal{E}_K\}$, and is represented by the signed distance field (SDF) induced by these extrusions. In our pipeline, $\{\mathcal{E}_1, \ldots, \mathcal{E}_i\}$ is recovered via point segmentation, while the current sketch $S_i$ is predicted with the improved transformer decoder.

Based on the above preliminaries, we present Point2Primitive, a direct sketch primitives prediction method for CAD reconstruction from point clouds. As shown in Figure 1, our method first segments points into extrusions, classifies base/barrel points, and then applies an improved transformer to predict curve types and parameters as a set-to-set problem. The segmentation head builds on PointCNN with fixed grid points, while the transformer refines features and enhances prediction accuracy.

### 3.2 CAD DATASET GENERATION

In this paper, we propose to generate the training dataset based on the Signed Distance Field (SDF). An SDF $\mathtt{SDF}(p_{\text{eval}}, b)$ is a continuous function that, for a given spatial eval point $p_{\text{eval}_i}$, outputs the eval point's distance to the closest boundary defined by $b$, whose sign denotes whether the point is inside (negative) or outside (positive) of the watertight surface. The data generation pipeline can convert any CAD sequence saved in JSON files following the style of the Fusion360 Gallery dataset (Willis et al., 2021) to the training data pairs.

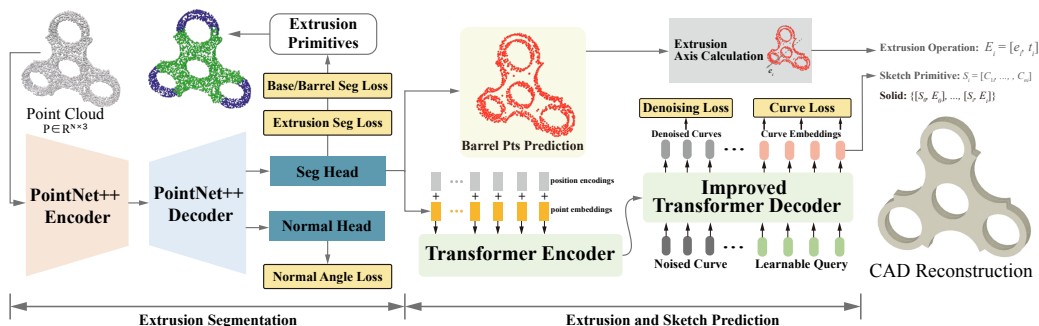

Figure 1: **Overview of Point2Primitive.** Our model consists of two main parts: extrusion segmentation and sketch prediction. In extrusion segmentation, a PointNet++ model is used to segment the barrel points and extrusions from the given point cloud to rebuild the topology. In sketch prediction, the point features from the Seg head are first refined using a vanilla transformer encoder and decoded into curves through the improved transformer decoder.

Given an extrusion point set $\mathbf{P}_{\mathcal{E}_i} \in R^{N_{\mathcal{E}_i} \times 3}$ and its sketch $S_i$. We first project the points $\mathbf{P}_{\mathcal{E}_i}$ to the sketch plane as eval points according to the transform matrix $W_i$ defined by its extrusion axis $e_i$. Then, its SDF is calculated to the boundary defined by the sketch $S_i$. The batch SDF calculation will be detailed in the Appendix F.1. The barrel label $b_i$ of each point can be defined as follows:

$$b_i = \text{Bool}(\text{SDF}(W_i \cdot \mathbf{P}_{\mathcal{E}_i}, S_i) \leq \text{thresh}) \tag{2}$$

where $\text{thresh}$ controls the label precision and $\text{thresh} = 0.01$ in this paper.

As for the Extrusion Label, each extrusion mesh is first built using pythonOCC (Paviot, 2022) by the CAD sequence. Then, the SDF of the input points to each extrusion boundary defined by its mesh is calculated. The extrusion label of each point is set as its closest extrusion mesh.

## 3.3 EXTRUSION SEGMENTATION

To decompose the raw point cloud into semantically meaningful extrusion bodies, we design an extrusion segmentation network that clusters points belonging to the same extrusion. As illustrated in Figure 1, the segmentation module is implemented using a PointNet++ encoder–decoder backbone (Qi et al., 2017), followed by two prediction heads.

Specifically, the encoder extracts hierarchical features from the input point cloud, while the decoder progressively recovers point-wise embeddings. On top of these embeddings, two parallel branches are employed: the *Normal Head* predicts point-wise surface normals, and the *Extrusion Segmentation Head* predicts extrusion-related classes. The Ext Head outputs logits $\hat{\mathbf{M}}$ corresponding to $2K$ categories, where each extrusion is represented by two complementary channels. For the $j$-th extrusion, the $(2j)$-th channel encodes the extrusion body assignment, whereas the $(2j+1)$-th channel encodes its associated barrel points. In this way, extrusion logits $\hat{\mathbf{W}}$ (for generating the extrusion $\mathbf{p}_k$) and barrel logits $\hat{\mathbf{B}}$ (for defining the the boundary $b$) can be computed as:

$$\hat{\mathbf{W}}_{:,j} = \hat{\mathbf{M}}_{:,2j} + \hat{\mathbf{M}}_{:,2j+1}, \tag{3}$$

$$\hat{\mathbf{B}}_{:,0} = \sum_j \hat{\mathbf{M}}_{:,2j}, \quad \text{and} \quad \hat{\mathbf{B}}_{:,1} = \sum_j \hat{\mathbf{M}}_{:,2j+1}. \tag{4}$$

Through this design, the network jointly captures extrusion topology and barrel boundary information, which facilitates accurate decomposition of the complex shape into extrusion units. Please refer to Appendix E for more implementation detail.

## 3.4 IMPROVED TRANSFORMER DECODER

To find the implicit design features encoded in the point features, we propose an improved transformer decoder to convert the refined point features into curves and directly predict the parameters. As shown in Figure 2, the curve embeddings encode the type information, and the curve parameters

are formulated as positional queries. By decomposing the learnable queries into type and parameter embeddings, explicit geometric priors are introduced, avoiding the abstract learning process and leading to high accuracy.

**Parameter as Position Encoding.** The curve parameters are formulated in a center-prior format detailed in Section 3.1, so the curve parameters can be formulated as position encoding $\mathbf{PE}_i$ to guide the attention. Given the curve parameters $\rho^i_j$ as the parameters of the $j$-th curve in the $i$-th sketch $S_i$. The corresponding positional queries $\mathbf{PE}^i_j$ are generated by:

$$\mathbf{PE}^i_j = \texttt{MLP}(f_{\text{pe}}(\rho^i_j)) = \texttt{MLP}(f_{\text{pe}}(x^i_{m_j}, y^i_{m_j})), \tag{5}$$

where $f_{pe}$ is the function that generates the sinusoidal embeddings from the curve center. Multi-layer perception (MLP) and ReLU activations are used to produce the position encodings $\mathbf{PE}^i_j$.

**Direct Curve Parameter Prediction.** The curve prediction procedure can be seen as feeding the positional queries (curve parameters) and curve embeddings (type queries) into the decoder to probe the curves $c^i_j$ that correspond to the position encodings while having similar patterns with the content in the point features. Furthermore, we propose to predict the curve

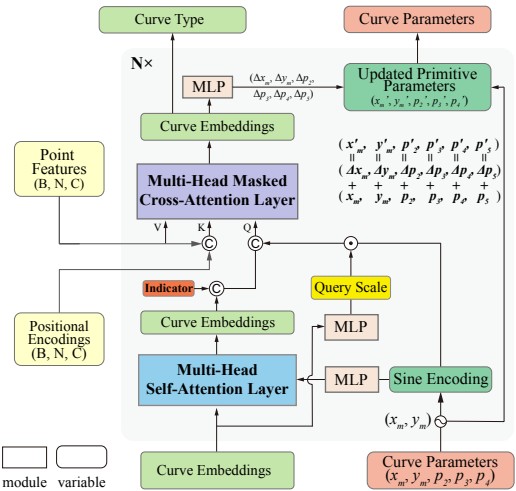

Figure 2: **The improved Transformer decoder.** The curve parameters are formulated as the position queries, while the curve embeddings encode the curve type. The curve parameters are updated in each Transformer layer through the predicted $\delta$.

parameters directly. The curve parameters are updated layer-by-layer in an autoregressive way to fit the target, as shown in Figure 2. The layer-by-layer update is based on the decoder output as follows:

$$\delta x^i_{m_j}, \delta y^i_{m_j}, \delta p^i_{2_j}, \delta p^i_{3_j}, \delta p^i_{4_j}, \delta p^i_{5_j} = \texttt{MLP}(O_n), \tag{6}$$

where $O_n$ is the output of the $n$-th decoder layer. The curve parameters are updated to $[x^{i'}_{m_j}, y^{i'}_{m_j}, p^{i'}_{2_j}, p^{i'}_{3_j}, p^{i'}_{4_j}, p^{i'}_{5_j}]$ in each layer, as shown in Figure 2. All intermediate parameter outputs are supervised by ground truth. We further utilize query denoising (Li et al.) to accelerate the convergence.

### 3.5 TRAINING OBJECTIVE

The Point2Primitive is trained using a multi-task objective composed of extrusion segmentation loss $\mathcal{L}_{\text{ext}}$, base-barrel classification loss $\mathcal{L}_{\text{bb}}$, normal $\mathcal{L}_{\text{norm}}$ and sketch $\mathcal{L}_{\text{skh}}$ prediction losses:

$$\mathcal{L} = \mathcal{L}_{\text{ext}} + \mathcal{L}_{\text{bb}} + \mathcal{L}_{\text{norm}} + \mathcal{L}_{\text{skh}}, \tag{7}$$

where the cross entropy loss is used in $\mathcal{L}_{\text{ext}}$ and $\mathcal{L}_{\text{bb}}$ while L-1 loss is used in $\mathcal{L}_{\text{norm}}$. The sketch prediction loss $\mathcal{L}_{\text{skh}}$ can be formulated as:

$$\mathcal{L}_{skh} = \sum_{i=1}^{N_C} \mathcal{L}_{\text{focal}}(\hat{t}_i, t_i) + \beta \sum_{i=1}^{N_C} \sum_{j=1}^{N_P} \mathcal{L}_{\text{p}}(\hat{\rho}_{i,j}, \rho_{i,j}), \tag{8}$$

$$\mathcal{L}_p = \mathcal{L}_1(\hat{\rho}_{i,j}, \rho_{i,j}) + \mathcal{L}_1(\hat{\rho}_{i,j}, \rho_{i,j}),$$

where $\mathcal{L}_{\text{focal}}(\cdot, \cdot)$ represents the focal loss (Lin et al., 2017) and $\mathcal{L}_1(\cdot, \cdot)$ and $\mathcal{L}_2(\cdot, \cdot)$ represents the L1 loss and L2 loss, respectively. Both denoising loss and curve Loss use the sketch prediction loss $\mathcal{L}_{skh}$, as shown in Figure 1. More specifically, in the L2 loss, we randomly select $n$ points on each curve and compute the chamfer distance as the L2 loss. $N_P$ is the number of parameters ($N_p = 6$ is this paper) while $N_C = 30$ is the number of curves. $\beta$ is the weight to balance both terms ($\beta = 2$ in this paper) and $\hat{\cdot}$ denotes the ground truth value.

| Methods | DeepCAD | | | | | | | Fusion 360 Gallery | | | | | | |
|---|---|---|---|---|---|---|---|---|---|---|---|---|---|---|
| | $Acc_t^{SKH}$ | $Acc_p^{SKH}$ | $CD$ | $ECD$ | $NC$ | IR | $\#\Delta P$ | $Acc_t^{SKH}$ | $Acc_p^{SKH}$ | $CD$ | $ECD$ | $NC$ | IR | $\#\Delta P$ |
| *Methods of Representing Sketch with Implicit SDF Encoding* | | | | | | | | | | | | | | |
| ExtrudeNet | 34.58% | 31.71% | 0.379 | 0.962 | 0.849 | 24.19% | 34.34 | 37.81% | 34.39% | 0.671 | 0.808 | 0.809 | 23.83% | 27.47 |
| Point2Cyl | 41.37% | 39.41% | 0.489 | 1.027 | 0.819 | 3.91% | 26.18 | 42.98% | 41.16% | 0.529 | 0.769 | 0.769 | 3.87% | 26.96 |
| SECAD-Net | 45.91% | 41.37% | 0.341 | 0.868 | 0.861 | 8.03% | 32.78 | 46.82% | 42.97% | 0.449 | 0.684 | 0.813 | 7.82% | 28.61 |
| *Generation Methods Based on Language Model* | | | | | | | | | | | | | | |
| DeepCAD | 82.61% | 73.36% | 0.898 | 1.883 | 0.823 | 14.12% | 5.89 | 77.31% | 69.81% | 7.128 | 8.729 | 0.719 | 13.18% | 7.47 |
| HNC-CAD | 84.31% | 76.71% | 0.827 | 1.064 | 0.846 | 6.01% | 4.43 | 80.62% | 72.19% | 4.381 | 5.571 | 0.748 | 5.92% | 6.14 |
| *Methods Based on Primitive Fitting* | | | | | | | | | | | | | | |
| UCSG-Net | - | - | 1.849 | 1.174 | 0.820 | - | 14.19 | - | - | 0.952 | 1.277 | 0.770 | - | 11.17 |
| CSG-Stump | - | - | 3.031 | 0.755 | 0.828 | - | 19.46 | - | - | 0.781 | 0.991 | 0.744 | - | 13.08 |
| Ours | **96.14%** | **86.81%** | 0.312 | 0.581 | 0.897 | 3.71% | 4.14 | **94.17%** | **83.52%** | 0.392 | 0.571 | 0.839 | 3.62% | 5.17 |

Table 1: **Evaluation results** on the test set of the CAD reconstruction dataset. CAD reconstruction and generation methods are selected for comparison.

## 4 EXPERIMENTS

### 4.1 EXPERIMENTAL SETUPS

**Datasets.** To evaluate the proposed model, we conduct experiments on DeepCAD (Wu et al., 2021) and Fusion 360 Gallery (Willis et al., 2021). The training data is generated as described in Sec. 3.2.

**Implementation Details.** We use PointNet++ (Qi et al., 2017) for extrusion segmentation. The Point2Primitive is trained for 400 epochs with a total batch size of 64 and learning rate (lr) $1e-5$ with linear warmup and step reduce lr scheduler. As for the query denoising settings, the noise rate for the curve type and parameters is 0.2 and 0.3, respectively, with 5 denoising groups. The transformer encoder and decoder contain 6 and 8 transformer layers, respectively, with eight attention heads and latent model dimension 256.

**Evaluation Metrics.** We adopt command type accuracy ($Acc_t^{SKH}$) and parameter accuracy ($Acc_p^{SKH}$) for quantitative evaluations of the sketch prediction following (González-Lluch et al., 2021). For networks that represent sketch by the implicit nueral field, the tools for convert sketch SDF to command are developed based on the tool by SECAD (Li et al., 2023a). Furthermore, following Chen and Zhang (2019), we also report chamfer distance ($CD$), edge chamfer distance ($ECD$), normal consistency ($NC$), and the number of generated primitives ($\Delta\#P$) to measure the quality of the recovered 3D geometry, more details of the quantitative metrics are shown in the Appendix B.

### 4.2 MAIN RESULTS

To demonstrate its effectiveness, we compare the proposed model to multiple existing methods, covering the methods of representing sketch with SDF encoding (Li et al., 2023a; Ren et al., 2022; Uy et al., 2022), the generation methods based on language model (Wu et al., 2021; Xu et al., 2023), and methods based on primitive fitting (Kania et al., 2020; Ren et al., 2021). The direct reconstruction results are visualized for comparison.

Table 1 summarizes the main results, and we can find that sketch curve type and parameter accuracy of the Point2Primitive are much higher than the other methods while the CD metric of the Point2Primitive is lower. The smallest CD and ECD value indicates that the presented CAD reconstruction method achieves better geometry fidelity while preserving accurate and sharp shape edges. This shows that by directly recovering curves of the extrusion primitive, our method can achieve better performance than the other extrusion-segmentation methods that represent the sketch with an implicit field. As for the number of generated primitives, our method is closer to human designs (The smallest $\#\Delta P$ value). This improvement is due to the explicit fine-grained reconstruction from each extrusion primitive; every parameter of the curves and extrusion operations are all predicted and optimized.

### 4.3 VISUALIZATION RESULTS

Figure 3 demonstrates the visualized reconstruction results on the DeepCAD and Fusion 360 Gallery datasets, respectively. It can be seen that the results of our method are more compact and complete, while the edges are sharp and sketches are accurate. This shows that our method can produce CAD reconstruction of high geometrical fidelity. The methods based on the language model can also

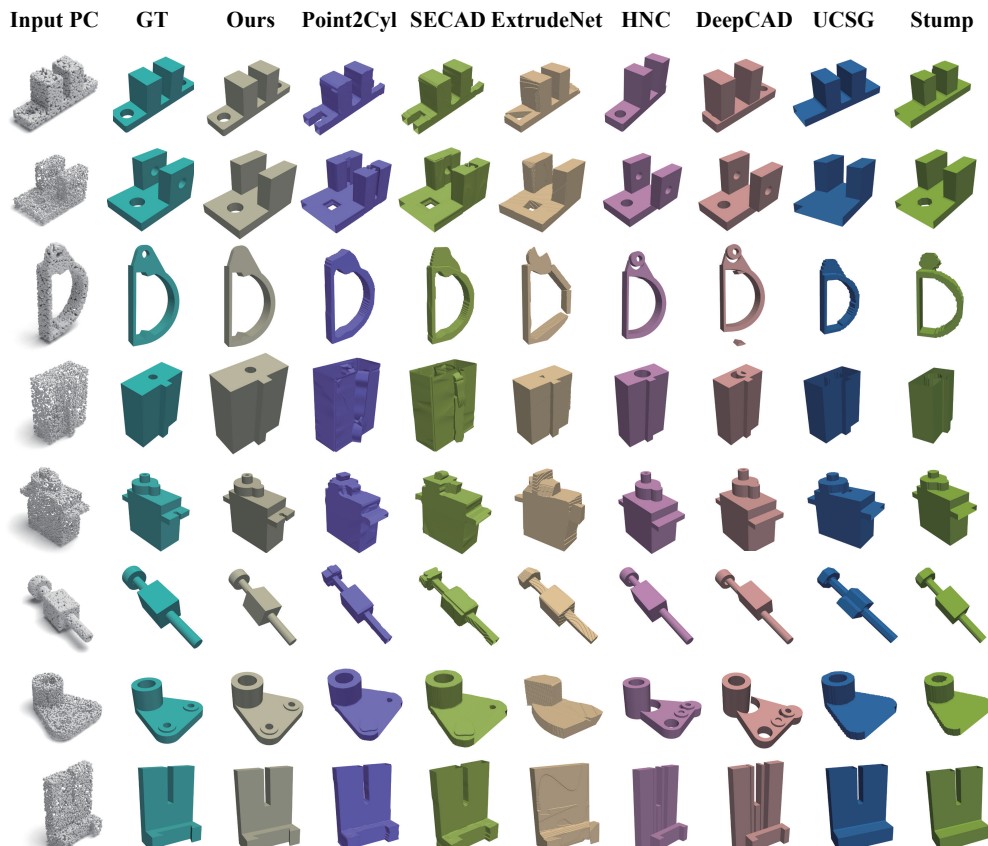

Figure 3: **Visual comparison** between reconstruction results on the DeepCAD (*the first 4 lines*) and Fusion360 gallery dataset (*the last 4 lines*).

| Methods | $Acc_t^{SKH}$ | $Acc_p^{SKH}$ | $CD$ | $ECD$ | $NC$ | $\#\Delta P$ |
|---|---|---|---|---|---|---|
| *Methods of Representing Sketch with Implicit SDF Encoding* | | | | | | |
| ExtrudeNet | 28.39% | 25.61% | 0.614 | 1.117 | 0.776 | 36.14 |
| Point2Cyl | 34.47% | 30.25% | 0.518 | 1.065 | 0.791 | 27.96 |
| SECAD-Net | 36.61% | 31.43% | 0.437 | 1.079 | 0.806 | 34.18 |
| *Generation Method Based on Language Model* | | | | | | |
| DeepCAD | 61.41% | 43.71% | 5.919 | 6.883 | 0.708 | 7.16 |
| HNC-CAD | 63.39% | 53.29% | 6.864 | 7.064 | 0.711 | 6.56 |
| *Methods Based on Primitive Fitting* | | | | | | |
| UCSG-Net | - | - | 2.146 | 1.273 | 0.797 | 14.81 |
| CSG-Stump | - | - | 3.681 | 0.958 | 0.804 | 20.16 |
| **Ours** | **94.67%** | **83.81%** | 0.410 | 0.607 | 0.819 | 4.97 |

Table 2: **Evaluation results** on the Augmented DeepCAD dataset.

produce accurate geometry structure, but the error of the parameters in the sequence results in some models of poor compactness. The primitive detection methods (USCG and Stump) fail to predict some holes in the model. Still, the edges are critically more accurate than the ExtrudeNet, whose reconstructions are less complete. Please refer to Appendix I for more visualizations.

## 4.4 COMPARISON ON AUGMENTED DEEPCAD DATASET

In addition to the experiments on the original DeepCAD and Fusion 360 Gallery datasets, we conducted extra comparisons on the augmented DeepCAD datasets. More specifically, we add random noise to the parameters of the CAD modeling sequence while keeping the types intact. This noise causes the augmented DeepCAD test set to have model structures similar to the original shape but with some modifications to the geometry details. We train each method on the original DeepCAD dataset and test all the methods on the augmented DeepCAD test sets.

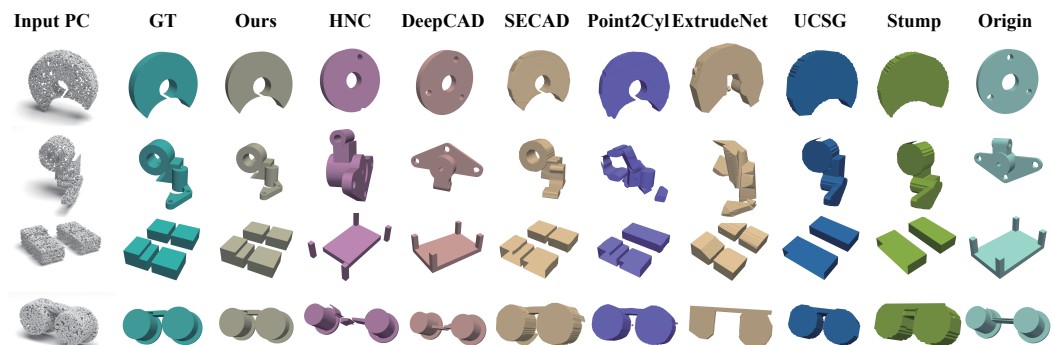

Figure 4: **Visual comparison** between reconstruction results on the augmented DeepCAD dataset.

| Metrics | Ours | | Point2Cyl | |
|---|---|---|---|---|
| | w/o $\mathcal{L}_{skh}$ | w/ | w/o $\mathcal{L}_{skh}$ | w/ |
| Ext. Seg IoU | 0.814 | 0.862 | 0.801 | 0.817 |
| BB. Seg Acc | 0.902 | 0.902 | 0.867 | 0.867 |
| EA. Angle Err ↓ | 7.416 | 7.173 | 8.391 | 8.267 |
| Fit Ext. ↓ | 0.0771 | 0.0527 | 0.0791 | 0.0741 |

(a) Quantitative comparison on DeepCAD.

(b) There constructed sketches by Point2Cyl and Point2Primitive.

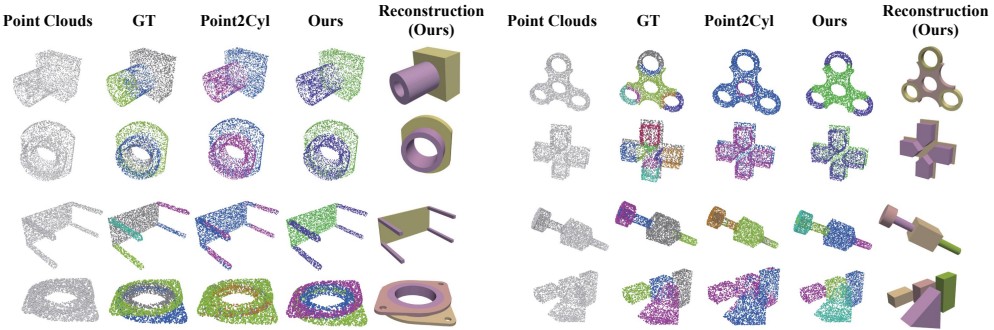

(c) The extrusion segmentation by Point2Cyl and Point2Primitive.

Figure 5: **Comparison with Point2Cyl.**

Table 2 demonstrates the quantitative results of the augmented DeepCAD dataset. We add the original shape (Origin) to the last column in the visualized results as shown in Figure 4. It can be seen that the methods based on the language model see a direct drop in the metrics of the sketch command type and command parameter, leading to a significant increase in the CD metric. Some reconstruction results by the language model preserve similarities to the original shape, leading to false reconstruction. Please refer to Appendix I for more visualizations.

The generated results are similar to the ground truth but with low geometry accuracies. On the other hand, reconstruction methods can still produce results that satisfy geometry fidelity. This implies that the methods based on the language model are less generalizable than the other methods. Also, both the quantitative metrics and the visual results verify that the proposed method can be generalized to reconstruction tasks beyond the train-test dataset.

## 4.5 DETAILED COMPARISON WITH POINT2CYL

In Table 5a, we present the quantitative comparison between our method and the most related approach, Point2Cyl. Compared with Point2Cyl, the metrics of the extrusion (Ext.) and barrel/based (BB.) segmentation are improved by 5.6% and 3.5%, respectively, leading to 13% and 32% reduc-

(a) Accuracy curve.

| No. | $\beta$ | SkhRep | ImpDec | $Acc_t^{SKH}$ | $Acc_p^{SKH}$ | $CD$ |
|---|---|---|---|---|---|---|
| 0 | 2.0 | - | - | 82.19% | 72.49% | 0.819 |
| 1 | 2.0 | ✓ | - | 85.71% | 78.61% | 0.674 |
| 2 | 1.0 | ✓ | ✓ | 93.16% | 84.91% | 0.341 |
| 3 | 3.0 | ✓ | ✓ | 93.24% | 84.73% | 0.337 |
| Ours | 2.0 | ✓ | ✓ | **96.14%** | **86.81%** | 0.312 |

(b) Quantitative results of ablation study.

Figure 6: **Ablation study**.

tion on the extrusion axis (EA.) angle and fitting (Fit Ext.) error, respectively. This shows that by supervising the extrusion segmentation with curve loss (type and parameter loss), more geometrical information is introduced, leading to better geometry accuracy.

We visualized the sketch predicted by Point2Cyl and our method, as shown in Figure 5b. It can be seen that because Point2Cyl represents the sketch by SDF, the edges of the sketch profiles are curved, while the counterpart of Point2Primitive is sharp and accurate. In addition, we present more visualizations of the predicted sketch by Point2Primitive in the Appendix C. Figure 5c (b) demonstrates the segmentation results by the two methods. It can be seen that even though the multiple parts of one extrusion are labeled with different extrusion labels, the segmentation network can still learn the geometry representation and cluster the points as the same extrusion. This implies that our method can predict simplified primitives that are closer to human design.

### 4.6 ABLATION STUDY

We perform ablation studies to carefully analyze the query denoising (DN), the improved transformer decoder, the sketch primitive definition, and the balancing weight in the loss function. All quantitative metrics are measured on the DeepCAD dataset.

**Query denoising.** It can be seen from the convergence curve shown in Figure 6a that by employing denoising in the training procedure, the ultimate sketch prediction accuracy is improved by 11.4% and 3.1% on $Acc_t^{SKH}$ and $Acc_p^{SKH}$, respectively. Please refer to Appendix. D for more visualizations. From the curves, one can see that by directly predicting the curve parameters and types, the accuracy is extremely high (type error below 0.5% and parameter error below 0.06).

**Decoder and Sketch Hierarchy.** We replace the improved decoder layer (ImpDec) with the vanilla transformer. Also, we utilize the sketch representation (SkhRep) method in Wu et al. (2021), which converts parameter prediction into a classification problem. As shown in Figure 6b, the $Acc_t^{SKH}$ improves by 12.2% by utilizing the ImpDec. Also, the center-prior sketch hierarchy contributes to an 8.4% improvement in $Acc_p$. This shows the effectiveness of the proposed improved transformer decoder.

**Balancing Weight.** Figure 6a shows that the balance weight $\beta = 2$ performs slightly better than the other conditions, and we set it to our default value.

## 5 CONCLUSION

The proposed Point2Primitive can produce CAD reconstruction of high geometric fidelity. The reconstruction is more accurate using curves as the sketch representation instead of SDF. The curve parameters are accurate by direct prediction in an autoregressive way through the improved transformer decoder. In further work, we plan to extend our approach to include more complex modeling commands.

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

# A  THE USE OF LARGE LANGUAGE MODELS (LLMS)

In this work, Large Language Models (LLMs) were used solely to polish the language for clarity and readability. No LLMs were employed for idea generation, experimental design, data analysis, or any other part of the research process.

# B  EVALUATION METRICS

## B.1  PRIMITIVE TYPE AND PARAMETER METRICS

The command type and command parameter accuracy evaluate how accurate the predicted command is. For each CAD command, we calculate the command type accuracy by:

$$ACC_{\text{type}} = \frac{1}{N_c} \sum_{i=1}^{N_c} \Psi[t_i = \hat{t}_i], \tag{9}$$

where $N_c$ denote the total number of the CAD commands, $C_i$ and $\hat{C}_i$ are the ground truth command type and predicted command type, respectively. $\Psi[i] \in \{0, 1\}$ is the indicator function. The command parameter accuracy is calculated by:

$$ACC_{\text{param}} = \frac{1}{K} \sum_{i=1}^{N_c} \sum_{j=1}^{|\hat{p}_i|} \Phi[|p_{i,j} - \hat{p}_{i,j}| < \epsilon] \Psi[C_i = \hat{C}_i], \tag{10}$$

where $K = \sum_{i=1}^{N_c} \Psi[C_i = \hat{C}_i]|p_i|$ is the total number of parameters in all correctly recovered commands. $p_{i,j}$ and $\hat{p}_{i,j}$ are ground-truth command parameters and predicted command parameters. $\epsilon$ is the tolerance threshold accounting for the parameter accuracy. In practice, we use $\epsilon = 0.01$.

## B.2  EXTRUSION SEGMENTATION IOU

To evaluate the extrusion segmentation, we use the Segmentation (Seg) IoU as the metric. The predicted extrusion segmentation labels are first reordered to correspond with the GT through Hungarian matches. The Seg IoU can be formulated as follows:

$$\text{Seg IoU} = \frac{1}{K} \sum_{k=1}^{K} RIoU(\mathbb{1}(\hat{\mathbf{W}}_{:,k}), \mathbf{W}_{:,k}), \tag{11}$$

$$RIoU(\mathbf{X}, \mathbf{Y}) = \frac{\mathbf{X}^\top \cdot \mathbf{Y}}{\|\mathbf{X}\|_1 + \|\mathbf{Y}\|_1 - \mathbf{X}^\top \cdot \mathbf{Y}}, \tag{12}$$

where $\hat{\mathbf{W}}_{:,k}$ and $\mathbf{W}_{:,k}$ are the predicted and ground-truth labels of the $k$-th extrusion, respectively. $\mathbb{1}(\cdot)$ indicates the one-hot conversion.

## B.3  BASE/BARREL POINT SEGMENTATION

The Base/Barrel (BB) point Segmentation accuracy is defined as follows:

$$\text{BB Acc} = \frac{1}{N} \sum_{i=1}^{N} (\mathbb{1}(\hat{\mathbf{B}}_{i,:}) == \mathbf{B}_{i,:}), \tag{13}$$

where $\hat{\mathbf{B}}_{i,:}$ and $\mathbf{B}_{i,:}$ are the predicted and ground-truth barrel label, respectively. $N$ is the number of input point clouds.

## B.4  EXTRUSION AXIS ANGLE ERROR

Extrusion axis (EA) angle error measures the angle error between the GT and prediction, which is defined as follows:

$$\text{EA Angle Error} = \frac{1}{K} \sum_{k=1}^{K} \arccos(\hat{\mathbf{e}}_{k,:}^\top \mathbf{e}_{k,:}), \tag{14}$$

where $\hat{\mathbf{e}}_{k,:}^\top$ and $\mathbf{e}_{k,:}$ are the predicted and ground-truth extrusion axis, respectively.

## B.5 EXTRUSION FITTING ERROR

The extrusion fitting (Fit Ext) error measures the average fitting error of each extrusion, which can be defined as follows:

$$\text{Fit Ext} = \frac{1}{K} \sum_{k=1}^{K} \mathcal{F}, \quad \text{where} \tag{15}$$

$$\mathcal{F} \triangleq \frac{1}{N_{\mathbf{P}_k}} \sum \left| \text{SDF}(\hat{\mathbf{s}}_k, \hat{\mathbf{S}}_k) - \text{SDF}(\mathbf{s}_k, \mathbf{S}_k)) \right|, \quad \text{and} \quad \hat{\mathbf{S}}_k = \prod (\mathbf{P}^{barrel_k}, \hat{e}_{k,:}, \hat{c}_{k,:}), \tag{16}$$

where $\hat{c}_{k,:}$ is the predicted extrusion center. $\prod(\cdot)$ projects per extrusion barrel points using the extrusion axis and center. The inner summation $\mathcal{F}(\mathbf{P}, k)$ represents the goodness of the $k$-th extrusion fitting.

## B.6 PRIMITIVE NUMBER

The primitive number is a metric of reconstructed sequence fidelity. We calculate the $\Delta \# P$ to measure the length difference between the reconstructed and ground-truth sequence, which is defined as follows:

$$\Delta \# P = \left| \hat{N}_c - N_c \right|, \tag{17}$$

where $\hat{N}_c$ and $N_c$ are the predicted and ground-truth primitive numbers.

## C VISUALIZED SKETCH PREDICTION

We provide more visualized results of the predicted sketch. Fig. 7 demonstrates some examples of the predicted sketch primitives. It can be seen that the predicted sketches are accurate both in curve type and parameter. However, some parameter errors can be seen in the Figure. Therefore, we develop a post-optimized method to eliminate the parameter errors, which is detailed in Section F.2.

## D VALIDATION LOSS W/O QUERY DENOISING

Except for the command type and accuracy curve, Fig. 8 demonstrates the validation loss of the proposed Point2Primitive during training. From the loss curves, we can see that the final loss of the curve type trained with DN is much smaller than the counterpart without DN (0.413 vs 1.1809). As for the curve parameter, the L-1 loss drops by 3.7%, and the distance loss drops by 4.9%. Also, the convergence time trained with DN is shorter than without DN.

## E THE IMPLEMENTATION OF THE EXTRUSION SEGMENTATION NETWORK

### E.1 EXTRUSION AND BARREL POINT SEGMENTAION

Figure 9 demonstrates the extrusion segmentation network. PointNet++ is used as the encoder and the decoder. The Normal head and Ext head produce the normal prediction $\hat{N}$ and the extrusion segmentation logits $\hat{\mathbf{M}}$. The extrusion segmentation logits $\hat{\mathbf{M}}$ produce the $2K-$classes of each point. The $(2k)$-th elements of each row of the $\hat{\mathbf{M}}$ encode the extrusion label, while the $(2k + 1)$-th elements encode the barrel label. Thus the extrusion $\hat{\mathbf{W}}$ and barrel $\hat{\mathbf{B}}$ segmentation logits are formulated as follows:

$$\hat{\mathbf{W}}_{:,j} = \hat{\mathbf{M}}_{:,2j} + \hat{\mathbf{M}}_{:,2j+1}, \tag{18}$$

$$\hat{\mathbf{B}}_{:,0} = \sum_j \hat{\mathbf{M}}_{:,2j}, \quad \text{and} \quad \hat{\mathbf{B}}_{:,1} = \sum_j \hat{\mathbf{M}}_{:,2j+1}. \tag{19}$$

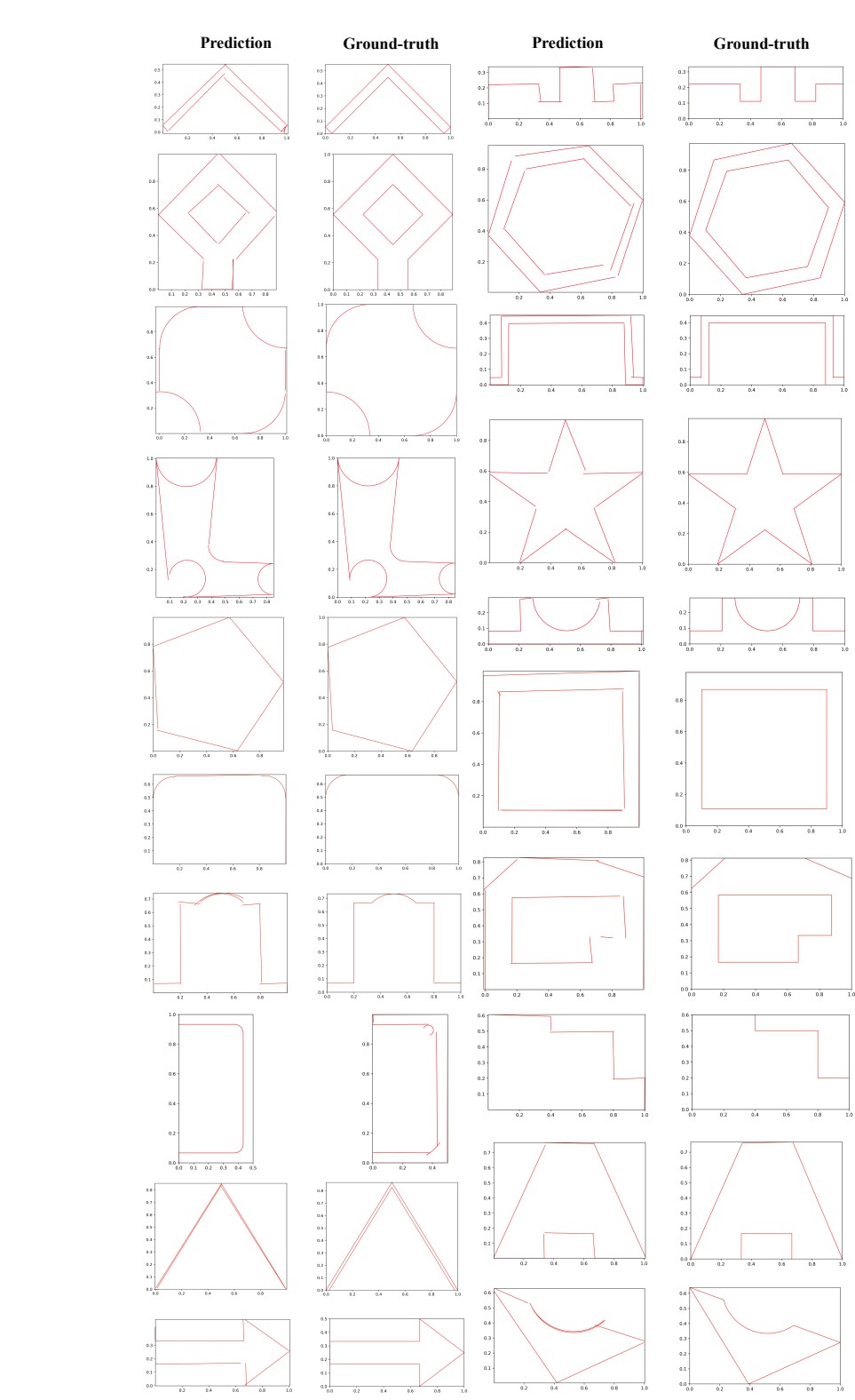

Figure 7: Visualization of the sketch primitive prediction.

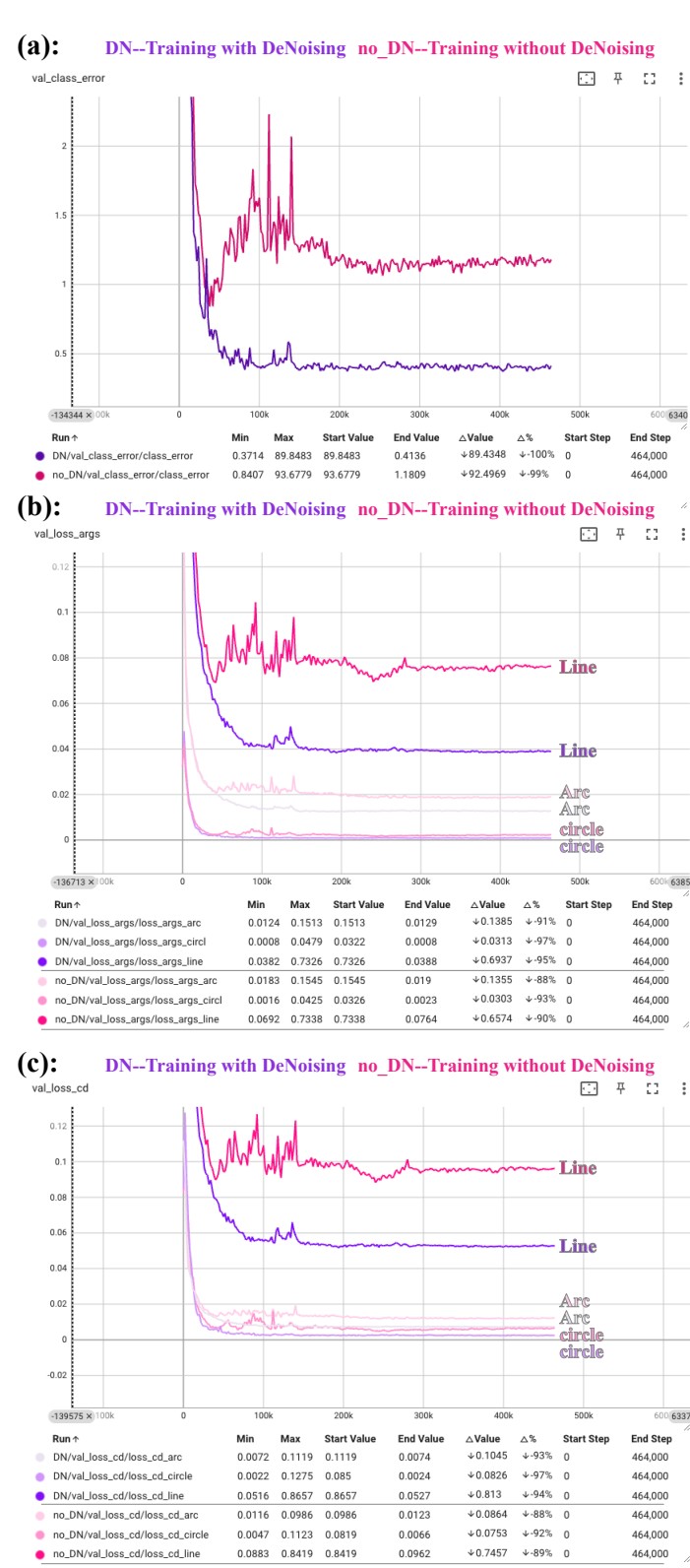

Figure 8: Validation curves with and without query denoising of the Point2Primitive during training: (a) the class error during training; (b) the L-1 loss of the primitive parameter during training; (c) the distance loss of the primitive parameter during training.

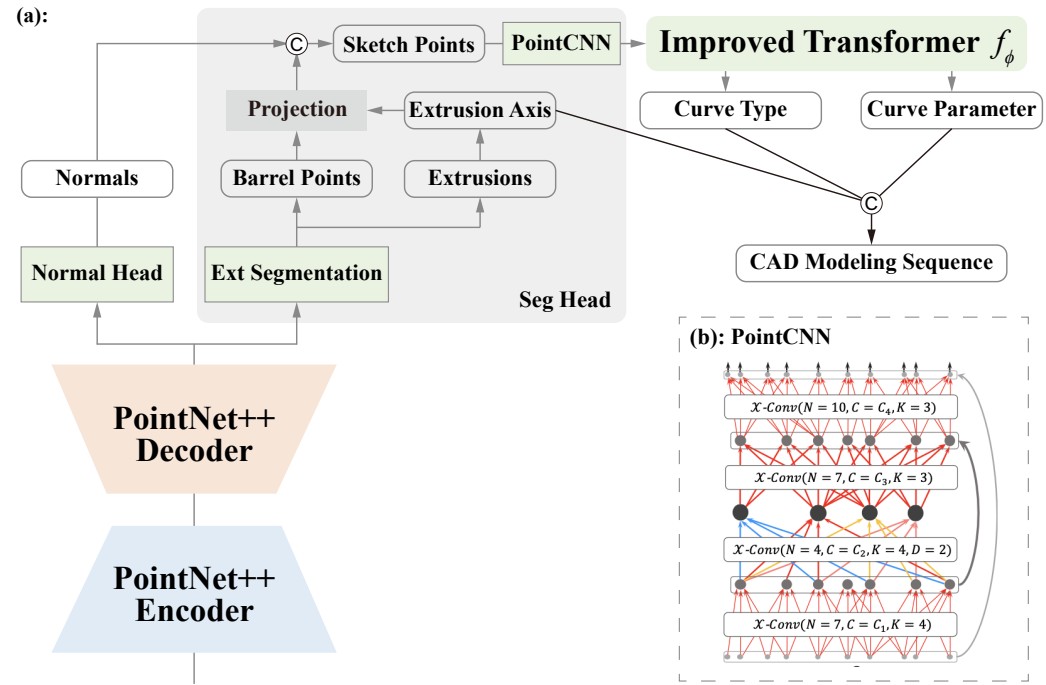

Figure 9: The implementation of the extrusion segmentation network: (a) the network architecture of the Point2Primitive; (b) the Sketch Point Encoder based on the PointCNN

### E.2 SKETCH POINT ENCODER

As shown in Figure 9, the sketch point encoder is developed based on the PointCNN(Li et al., 2018). Instead of using random or fps points, fixed points are set as the representation points. Also, to provide position encodings for the transformer, a learned position encoding is utilized.

## F POST OPTIMIZATION

### F.1 THE BATCH SDF COMPUTING

We develop a batch SDF computing method to calculate the sketch SDF considering batch input. More specifically, the curves of each sketch will be divided into three curve types $(L, A, C)$, and curves of the same type are calculated in batches. We now provide an anonymous GitHub repository to present the SDF computing in https://github.com/AnonymousRepo1234/Point2Primitive/blob/main/SDF_batch_cal.py.

### F.2 CURVE FINE-TUNING

To produce more accurate curve parameters for CAD reconstruction, we develop a post-optimization method for the predicted curve parameters, as shown in Figure 10. The predicted sketch points can be used to optimize the predicted curve parameters. First, we project the barrel point into sketch points. Then, we calculate the sketch SDF to the sketch points using batch SDF computing. The sketch SDF to its sketch points is set as the loss function. Thirdly, the curve parameters are set as model parameters and updated using the autograd in Pytorch supervised by the SDF. The sketch SDF to its sketch points is supposed to be zero. The algorithm is shown in Algorithm 1.

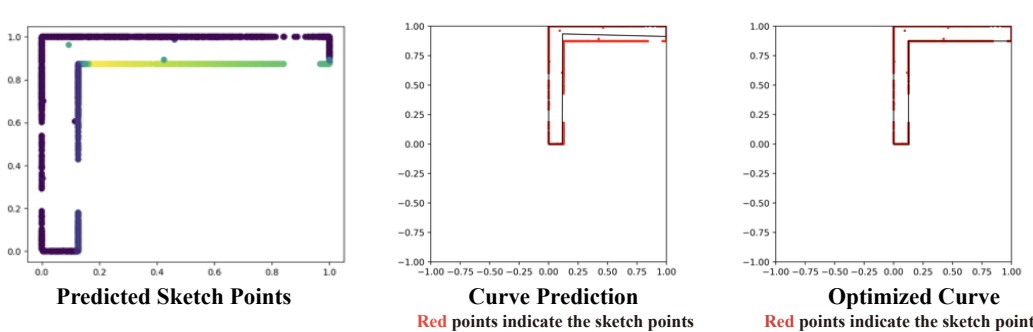

Figure 10: Post optimization of curve prediction.

---

**Algorithm 1** Curve Fine-tuning

---

**Input:** $Curves$
**Output:** $T$ as modified Curves
  $Curves_{norm} \leftarrow normalize(Curves)$
  $Curves_{norm} \leftarrow sort(Curves_{norm})$               ▷ sorted by starting point
  $S \leftarrow Curves_{norm}$
  $T \leftarrow \emptyset$                                ▷ as modified Curves
  **while** $S \neq \emptyset$ **do**
      $x \leftarrow S.front$
      **for** each curve $l \in S$ **do**             ▷ modify neighborhood line
          **if** $\|x.end, l.start\| < Threshold$ **then**
             $x.end, l.start \leftarrow mid(x.end, l.start)$
             $T \leftarrow T \cup \{x\}$
             $S \leftarrow S \backslash \{x\}$
             **break**
          **else if** $\|x.end, l.end\| < Threshold$ **then**
             $x.end, l.end \leftarrow mid(x.end, l.end)$
             $l \leftarrow reverse(l)$
             $T \leftarrow T \cup \{x\}$
             $S \leftarrow S \backslash \{x\}$
             **break**
          **end if**
      **end for**
  **end while**
  $T \leftarrow post\_process(T)$

---

### F.3 Loop Opmitzation

---
**Algorithm 2** Loop Opmitzation

---
**Input:** point_cloud, Curves
**Output:** params as modified sketch curve params
  eval_points ← in_pc(point_cloud)                       ▷ as pc to be formulated
  cmd, params ← Initialize(sketch)     ▷ cmd as mask of curve type, params as shape of sketch
  **for** each epoch in $epoches$ **do**
      line_loss ← SDF(eval_points, params[line_mask(cmd)])   ▷ calculate line, arc and circle loss
      arc_loss ← SDF(eval_points, params[arc_mask(cmd)])
      circle_loss ← SDF(eval_points, params[circle_mask(cmd)])
      loss ← concat(line_loss, arc_loss, circle_loss, -1)
      loss.min()                      ▷ minimize the last element of loss tuple
      loss.backward()
  **end for**

---

The sketch should contain loops, which are composed of curves connected end to end. To connect the predicted curves into loops, we use an optimization method based on Greedy Algorithm as shown in Algorithm 2.

## G Extrusion Parameters Calculation

We calculate the parameters of the extrusion operation by the obtained barrel points. The parameters of the extrusion operation $E_i = (\hat{e}_i, \hat{t}_i)$ consists of the extrusion axis $\hat{e}_i$, extrusion position $\hat{o}_i$, and extrusion extent $\hat{t}_i$. Firstly, the optimal extrusion axis of an input extrusion cylinder PC is given by $\hat{e}_i = \text{argmin}_{e_i, \|e_i\|=1}(e_i^T H_\phi e_i)$, where:

$$H_\phi = \mathbf{N}^\top \Phi_{barr}^\top \Phi_{barr} \mathbf{N} - \mathbf{N}^\top \Phi_{base}^\top \Phi_{base} \mathbf{N}, \tag{20}$$

where $\mathbf{N}$ denotes the normals of the input points. $\Phi_{barr} = diag(\phi_{barr})$, $\Phi_{base} = diag(\phi_{base})$. $\phi_{base}$ and $\phi_{barr}$ indicate the barrel and base weights (assigned to all points) predicted by the segmentation head, respectively. The solution is given by the eigenvector corresponding to the smallest eigenvalue of $H_\phi$.

The extrusion extent $\hat{t}$ is the point projection range on the extrusion axis, which can be calculated by Equation 21.

$$\hat{t}_{min} = \min_{P_i \in P_{barr}} (e_i \cdot (P_i - \hat{o})) \quad \text{and} \quad \hat{t}_{max} = \max_{P_i \in P_{barr}} (e_i \cdot (P_i - \hat{o})), \tag{21}$$

## H More Details of the Improved Transformer Decoder

### H.1 Command Parameter Noise

For each input training pair $(p_i, S_i)$, we add random noises to both their curve types and curve parameters. A random noise $(\Delta x, \Delta y)$ is added to the endpoint coordinates. The values of $(\Delta x, \Delta y)$ are limited between $[0.0, 0.2]$ so that the noised endpoint will not shift too far from the ground-truth position. The other primitive parameters $(x_{m_i}, y_{m_i}, r_i)$ are randomly sampled from $(\phi x_{m_i}, \phi y_{m_i}, \phi r_i), \phi \in [-1, 1]$.

For curve type noising, the GT command types are randomly flipped.

### H.2 Attention Mask

An attention mask is used to prevent information leakage. There are two reasons for this. Firstly, the matching part might be able to obtain information from the noised curve types and easily infer the target curve types. Secondly, the various noised groups of the same ground-truth curve might exchange information with each other.

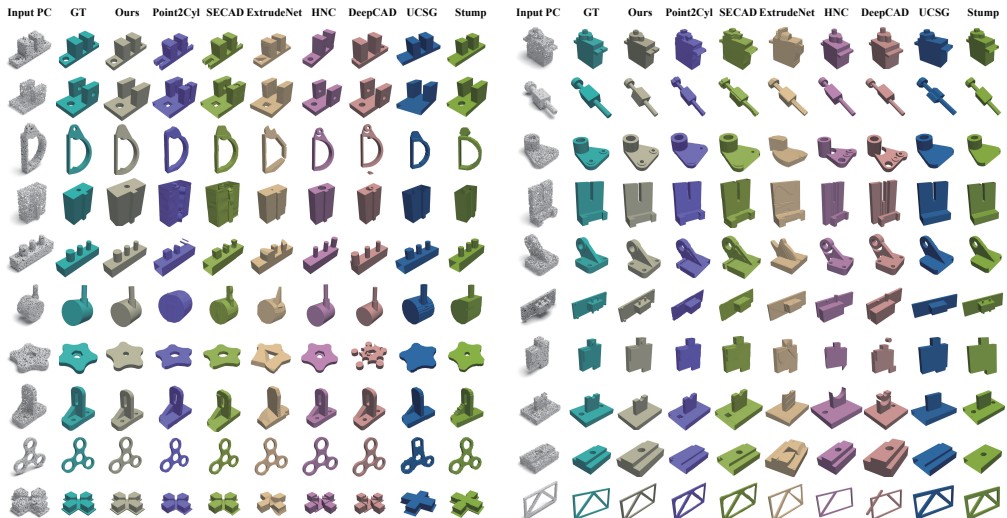

Figure 11: **More visual comparison** between reconstruction results on the DeepCAD and Fusion360 gallery dataset.

Given a sketch contains $N_c$ curves. The noised ground-truth curves of all the primitives are first divided into $K$ groups. The denoising part is then denoted as:

$$q = \{g_0, g_1, ..., g_{K-1}\}, \quad \text{and} \quad g_j = \{q_0^k, q_1^k, ..., q_{M-1}^k\} \tag{22}$$

where $g_k$ is the $k$-th denosing group. $q_m^k$ is the $m$-th query in the denoising group. Each denoising group contains $M$ queries where $M$ is the number of commands in one input PC.

The attention mask $A = [a_{ij}]_{W \times W}$ can then be formulated as:

$$a_{ij} = \begin{cases} 1, & \text{if } j < K \times M \text{ and } \lfloor \frac{i}{M} \rfloor \neq \lfloor \frac{j}{M} \rfloor; \\ 1, & \text{if } j < K \times M \text{ and } i \geq K \times M; \\ 0, & \text{else} \end{cases} , \tag{23}$$

where $a_{ij} = 1$ means the $i$-th query cannot see the $j$-th query and $a_{ij} = 0$ other wise.

The decoder embeddings are taken as curve embeddings in our model. Therefore, an indicator is appended to the curve embeddings to distinguish the denoising part from the matching part, as shown in Figure 2, where 1 means the query belongs to the denoising part and 0 means the query belongs to the matching part.

## I   MORE VISUALIZATIONS

We provide more visualization results as shown in Figure 11 and Figure 12.

## J   APPLICATIONS

In addition to the public CAD dataset, we also tested our method on some practical datasets. Figure 13 demonstrates some practical applications of the presented CAD reconstruction method.

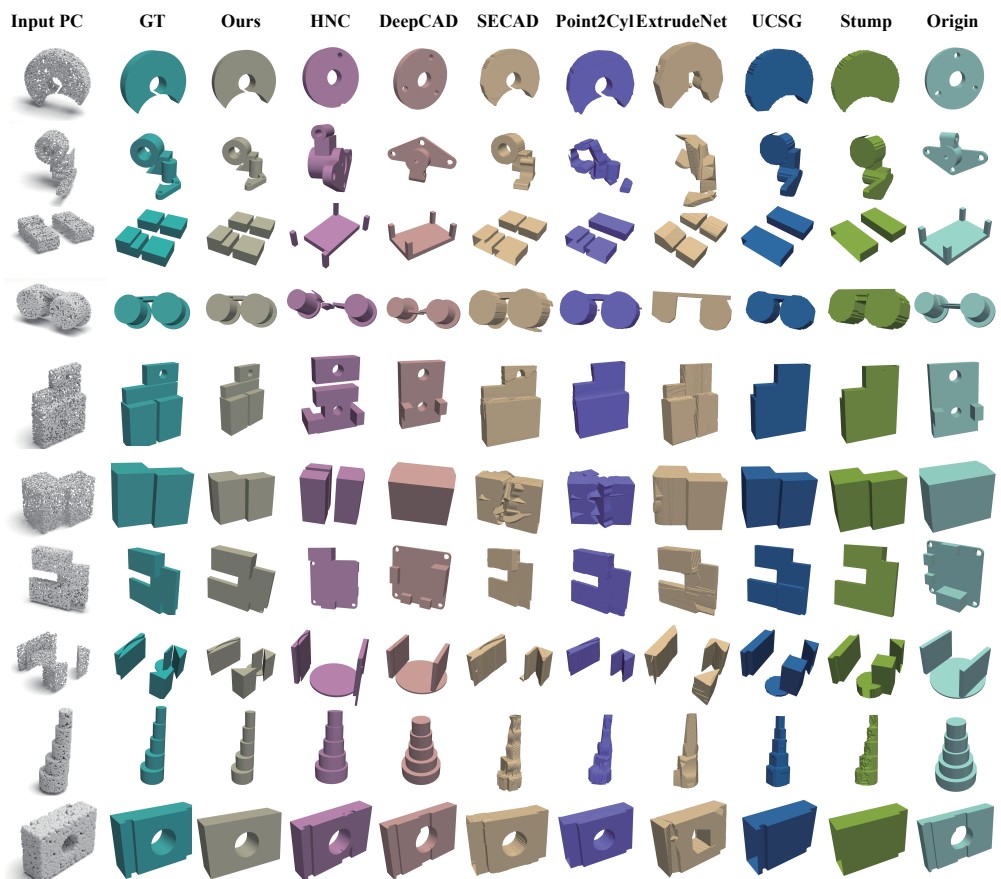

Figure 12: **More visual comparison** between reconstruction results on the augmented DeepCAD dataset.

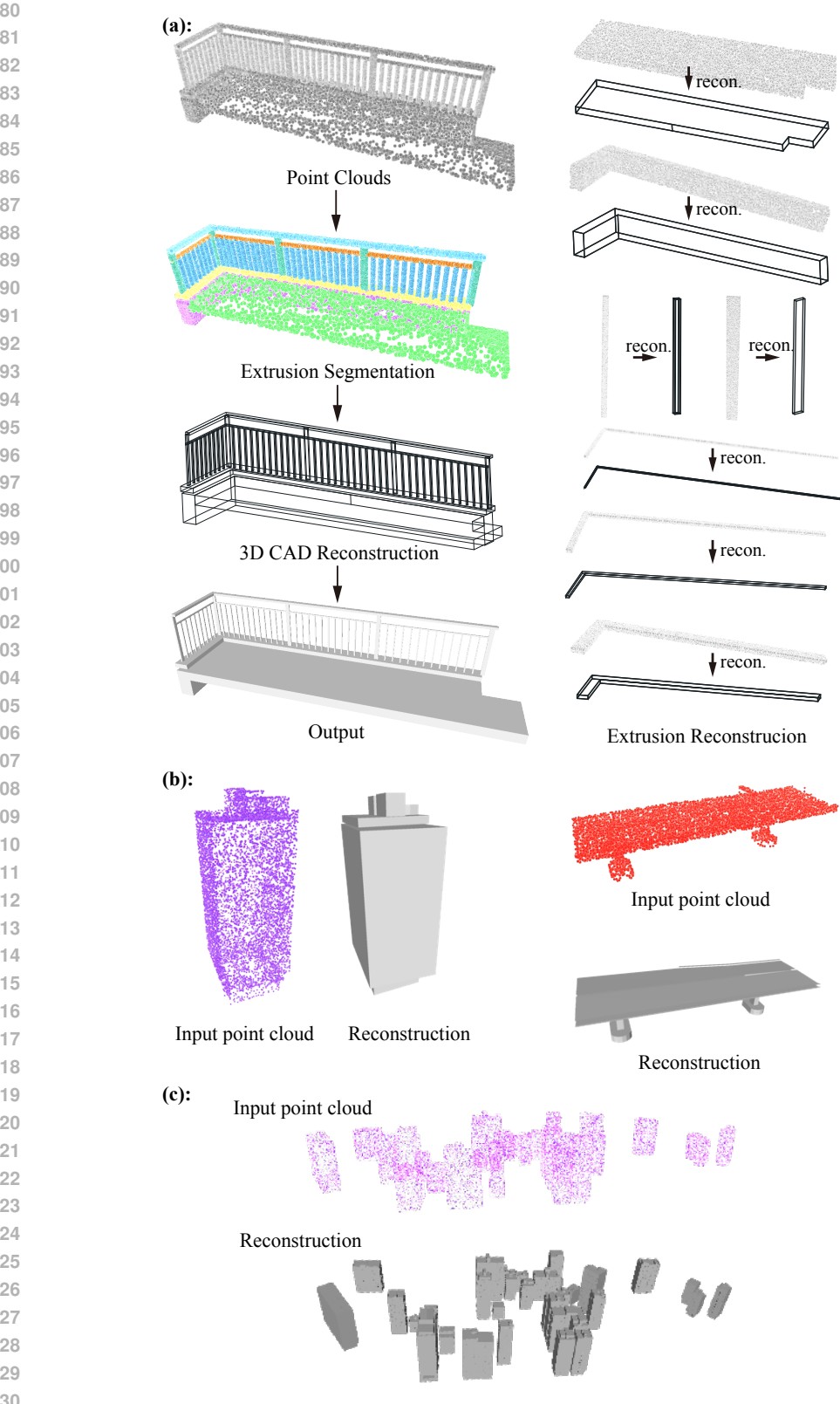

Figure 13: Examples of some practical application: (a) reconstruction of the structural components; (b) reconstruction of the buildings; (c) reconstruction of the small urban agglomeration.

