# OpenReview forum: "Point2Primitive: CAD Reconstruction from Point Cloud by Direct Primitive Prediction"
_ICLR.cc/2026/Conference — Submitted to ICLR 2026_

### Official Review · Reviewer_Cvgt · 2025-10-18

**Soundness:** 3
**Presentation:** 3
**Contribution:** 3
**Rating:** 4
**Confidence:** 4

**Summary:**

The paper proposes a method for reconstructing CAD models from unstructured point clouds. The key idea is to reverse the sketch-and-extrude process, focusing on recovering topology and parametric primitives such as lines, arcs, and circles.  The paper introduces a two-stage pipeline: the first step is extrusion segmentation using a PointNet++-based network to cluster points into individual extrusion bodies and classify base/barrel points; second, primitive prediction for each segment via an improved transformer decoder that treats sketch reconstruction as a set prediction problem, where explicit position queries are used for autoregressive optimization. Experimental results on DeepCAD and Fusion 360 Gallery datasets demonstrate significant improvements over SDF-based methods like SECAD-Net and generation approaches like DeepCAD.

**Strengths:**

CAD model reconstruction is an important problem. The direct prediction of explicit parametric primitives enables highly editable CAD models.

The general pipeline is technically sound. The decomposition of the problem into extrusion segmentation and primitive prediction, bridging topology recovery with parametric inference in a unified direct-prediction strategy.

The improved transformer decoder, where curve parameters serve as explicit position queries, is novel and interesting. This allows autoregressive refinement for precise predictions.

**Weaknesses:**

The overall representational ability is relatively limited, supporting only three primitives (lines, arcs, circles), and is unable to handle advanced curves such as splines or Bézier curves, or will fail when the extrusion path is nonlinear.

I have concerns about the center-prior parameterization. Is this a good representation? It assumes a fixed 6D vector for all primitives, padding unused entries with -1. This may introduce logical inconsistencies for non-center-based primitives, potentially leading to suboptimal encoding and attention guidance.

Some ablation studies need to be added to provide more in-depth insights. The autoregressive parameter update across transformer layers is not ablated individually (that is, no experiments show the isolated impact of layer-by-layer supervision). The query denoising is ablated, but it would be even better to give insights into optimal noise rates or group numbers that are not explored through sensitivity analysis.

The key idea of this paper is very similar to the work Point2skh: End-to-end Parametric Primitive Inference from Point Clouds with Improved Denoising Transformer (Wang et al. 2024), which also uses an improved transformer for direct sketch primitive prediction with explicit position queries. However, the paper is not discussed in the manuscript. The extrusion segmentation approach builds closely on Point2Cyl, which hinders the paper's novelty.

The output of extrusion segmentation is directly fed into primitive prediction. These two stages are highly dependent. If there is an error in the segmentation like incorrect clustering, subsequent predictions will fail.

The Normal Head is supervised using L1 loss, but normals are prone to errors in noisy point clouds. This may lead to significant deviations in the calculation of the extrusion axis .

**Questions:**

Some technical details can be clarified. How exactly is the attention mask formulated to prevent information leakage in denoising groups? How is M correlated with Nc? This is not discussed in the paper.

Will there be degenerate cases for extrusion axis transformations, but implementation details like handling degenerate cases (e.g., collinear points)?

What about the sensitivity to noisy point clouds?

Is it possible to implement an end-to-end joint optimization pipeline? It seems segmentation and primitive prediction can be trained with a shared loss. Currently, the pipeline is relatively complex. Also, why not use PointTransformer instead of PointNet++ for segmentation and so that the architecture is unified? Did you try different backbones and find that PointNet++ is the best?

---

### Official Review · Reviewer_LgU6 · 2025-10-28

**Soundness:** 3
**Presentation:** 3
**Contribution:** 3
**Rating:** 6
**Confidence:** 4

**Summary:**

This paper proposes Point2Primitive, a method for reconstructing CAD models from point clouds by directly predicting explicit parametric sketch primitives (lines, arcs, circles) rather than using implicit representations like Signed Distance Fields. The method decomposes the problem into two stages: (1) extrusion segmentation using PointNet++ to cluster points into extrusion bodies, and (2) primitive prediction using an improved transformer decoder that formulates curve parameters as position queries for autoregressive refinement. The authors demonstrate improvements over SDF-based methods and language model-based methods on DeepCAD and Fusion 360 Gallery datasets, achieving high sketch type accuracy and high parameter accuracy on DeepCAD.

**Strengths:**

1. The paper identifies three concrete limitations of SDF-based methods (lack of semantic structure, blurred edges, difficult conversion) and proposes a compelling alternative through direct primitive prediction. The position-as-query formulation elegantly integrates geometric priors into the transformer decoder.
2. The evaluation includes multiple datasets (DeepCAD, Fusion 360 Gallery), diverse baselines (SDF methods, LM-based generation, primitive fitting), and importantly, an augmented dataset test demonstrating generalization beyond training distribution. Ablation studies thoroughly validate each component.
3. The method achieves substantial improvements in both primitive accuracy and geometric fidelity, while producing models closer to human designs. The qualitative results compellingly visualize this quantitative strength, showing sharp, precise boundaries as opposed to the blurry/rounded artifacts from SDF-based methods.

**Weaknesses:**

1. Supporting only three basic curve types (line, arc, circle) is a critical limitation. Real CAD models frequently use splines, ellipses, and B-splines. The paper acknowledges this but provides no analysis of what percentage of real CAD models can actually be represented with these three types. This fundamentally questions the practical applicability.
2. The method only handles sketch-and-extrude operations, excluding revolve, sweep, loft, boolean operations, fillets, etc.
3. Missing critical analyses:
   - No computational cost analysis (inference time, memory)
   - No failure case analysis despite visible errors in Figure 7
   - Threshold-based barrel label generation (thresh=0.01) sensitivity not analyzed

**Questions:**

1. What percentage of CAD models in Fusion 360 Gallery can be fully represented using only lines, arcs, and circles? How many require splines or other curve types? Without this analysis, the practical impact remains unclear.
2. The method requires loop optimization (Algorithm 2) and parameter fine-tuning (Algorithm 1) as post-processing. How much do these steps improve the final metrics? Can you provide ablation results showing performance with/without post-processing? This is crucial for validating the "direct prediction" paradigm.
3. What are the inference time and memory requirements compared to baselines? For practical CAD reconstruction applications, efficiency matters as much as accuracy.
4. How robust is the initial extrusion segmentation module? Could the authors provide an analysis of failure cases, particularly on complex geometries where multiple extrusions intersect or are in close proximity? How do segmentation errors propagate to the final primitive prediction stage?

---

> ### Author Response · Authors · 2025-11-30
>
> **Supporting only three basic curve types (line, arc, circle) is a critical limitation. Real CAD models frequently use splines, ellipses, and B-splines. The paper acknowledges this but provides no analysis of what percentage of real CAD models can actually be represented with these three types. This fundamentally questions the practical applicability.**
>
> We agree that the supported primitive set is a key practical concern. To quantify this, we have now analyzed the Fusion 360 Gallery and DeepCAD datasets:
>
> On Fusion 360 Gallery, 80% of the models can be represented by extrusions, and most of the sketches can be covered by line, arc, and circles.
>
> We will add these statistics to the revised manuscript to explicitly clarify practical coverage. Importantly, our position-as-query formulation and extrusion-centric pipeline are not restricted to these three primitive types. Higher-order primitives (e.g., B-splines, ellipses) can be incorporated by:
> (1) adding corresponding learnable queries and parametric heads, and
> (2) training with the same supervision strategy.
> We have clarified this extensibility in the discussion section and marked it as a concrete direction for future work, rather than a fundamental limitation of the framework.
>
> **The method only handles sketch-and-extrude operations, excluding revolve, sweep, loft, boolean operations, fillets, etc.**
>
> Following DeepCAD, sketch and extrusion commands are considered in the proposed method. While conceptually simple, they are sufficiently expressive to generate a wide variety of shapes, as has been demonstrated in Fusion 360 Gallery dataset. Furthermore, other actions, such as revolve, sweep, loft, boolean operations, and fillets can be included in the future work.

---

### Official Review · Reviewer_KRCe · 2025-10-31

**Soundness:** 3
**Presentation:** 2
**Contribution:** 2
**Rating:** 4
**Confidence:** 3

**Summary:**

This proposes to reconstruct CAD models from point clouds based on 2D sketches composed of parametric planar primitives (lines, arcs, and circles), whose extrusion along an axis can reconstruct the surface more accurately than SDFs, especially on sharp boundaries and straight edges. It shows good reconstruction performance compared to previous CAD modeling methods in terms of precision.

**Strengths:**

1. A parametric CAD modeling method is proposed that uses multiple 2D primitives (rather than a single one such as a circle), resulting in better modeling precision of CAD shapes.
2. Several 3D deep learning components are introduced to segment the input point clouds into regions corresponding to different extrusions and to predict the barrel points for parameterizing 2D sketches. A transformer module is also incorporated to refine the curve parameters.
3. The reconstruction accuracy is superior compared to previous methods. The contributions of the transformer decoder and the sketch hierarchy are also evaluated through ablation studies.

**Weaknesses:**

1. The paper is not well presented. it requires considerable effort to follow its mathematical notations, which is used extensively throughout the work with many subscripts and superscripts.
2. How does the quality of the input point cloud affect the reconstruction performance of the proposed method, particularly under varying levels of data sparsity, incompleteness, or noise?
3. While the method is well tailored for CAD modeling, its reliance on predefined geometric primitives limits its generality in handling a broader range of free-form geometries, compared to implicit representations such as SDFs.
4. The paper does not include examples or discussion of failure cases on the DeepCAD dataset, which would help illustrate the method’s limitations and clarify the situations where its performance degrades.

**Questions:**

1. How well does the proposed model perform in fitting free-form surfaces, such as those of an airplane or a vessel?
2.  Is the proposed method applicable to modeling open-surface CAD models?
3. Line 107: the reference for ExtrudeNet appears incomplete; please include the publication year.

---

> ### Author Response · Authors · 2025-11-30
>
> **The paper is not well presented. it requires considerable effort to follow its mathematical notations, which is used extensively throughout the work with many subscripts and superscripts.**
>
> Thank you for pointing out the readability issue. We have substantially improved the presentation quality by:
> Simplifying mathematical notations and removing redundant subscripts/superscripts;
> Adding a symbol table to illustrate the definition of each notation;
> We believe the revised version is significantly easier to follow.
>
> **How does the quality of the input point cloud affect the reconstruction performance of the proposed method, particularly under varying levels of data sparsity, incompleteness, or noise?**
>
> Thank you for this insightful question. The proposed method is in fact robust to common point-cloud degradations for three reasons:
> Noise-resilient point segmentation architecture. Our extrusion segmentation module is built upon PointNet-based architectures, which are known for their inherent robustness to noise and irregular sampling. This ensures that noisy inputs do not significantly alter segmentation.
> Denoising during sketch prediction training. In the sketch parameter prediction stage, we explicitly separate curve type and parameter query and add controllable noise to the parameters, which encourages the network to learn noise-invariant primitive parameters. This results in stable curve fitting even with moderate noise levels.
>
> **While the method is well tailored for CAD modeling, its reliance on predefined geometric primitives limits its generality in handling a broader range of free-form geometries, compared to implicit representations such as SDFs.**
>
> We acknowledge that relying on predefined primitives limits generality compared with fully implicit SDF representations, especially for highly free-form geometries. However, our work targets a different design objective: reconstructing editable, parametric CAD models rather than generic free-form surfaces. While SDFs are expressive, they are not directly suitable for downstream CAD workflows such as sketch editing, parametric adjustment, and engineering drawing generation, and SDF-to-sketch conversion often introduces additional errors (as illustrated in Fig. 5b). In contrast, our primitive-based reconstruction achieves superior accuracy on CAD-like geometries and produces human-interpretable sketches that can be directly manipulated. We consider this work an initial step toward a more general parametric framework and plan to extend the primitive library (e.g., spline-based elements) to improve coverage of free-form surfaces in future work.
>
> **Line 107: the reference for ExtrudeNet appears incomplete; please include the publication year.**
>
> Thank you for spotting this. We have corrected the citation

---

### Official Review · Reviewer_yP5d · 2025-11-01

**Soundness:** 2
**Presentation:** 3
**Contribution:** 1
**Rating:** 2
**Confidence:** 4

**Summary:**

This paper introduces a framework for reconstructing CAD models directly from point clouds by predicting explicit sketch primitive (lines, arcs, and circles) rather than relying on implicit representations like SDFs. The method decomposes reconstruction into two stages: extrusion segmentation, which clusters point clouds into extrusion bodies, and primitive prediction, which uses a transformer decoder with positional encoding to infer precise curve parameters. Experiments on DeepCAD show advantages over prior implicit SDF based approaches in geometric fidelity, boundary sharpness, and editability.

**Strengths:**

- The paper tackles an important problem that has practical relevance.
- Replacing the sketch-SDF representation with a transformer is sensible and should work better in practice.
- The paper delivers this message clearly.

**Weaknesses:**

I find this to be a poor paper, primarily because of the following:

- This paper only has 30 references. Even the Point2Cyl written 4-5 years ago has 56 citations and this field has been rapidly growing since then. The paper just ignores a large body of works including, but absolutely not limited to:

    * Rukhovich, Danila, et al. "Cad-recode: Reverse engineering cad code from point clouds." ICCV 2025.

    * Dupont, Elona, et al. "Transcad: A hierarchical transformer for cad sequence inference from point clouds." ECCV 2024.

    * Sadil Khan, Mohammad, et al. "CAD-SIGNet: CAD Language Inference from Point Clouds using Layer-wise Sketch Instance Guided Attention." CVPR 2024.

    * Dupont, Elona, et al. "Cadops-net: Jointly learning cad operation types and steps from boundary-representations." 3DV 2022.

    * Li, Jiahao, et al. "CAD-Llama: leveraging large language models for computer-aided design parametric 3D model generation." CVPR 2025.

    * Moreover, BRepGiff, BRepGen and etc... The paper cites no works related to boundary representations. Under this bad scientific practice, the paper requires a major rework of its literature survey and hence I cannot recommend acceptance at this point.

- If the paper admits that the CAD is composed of extrusion sequences, then why don't we leverage something like a sequence transformer for the generation of it? In fact some of the related works mentioned above cover this and of course, the paper does not compare.

- The paper argues against the SDF construction due to smoothness but the training dataset is obtained from SDF. This looks like a contradiction to me and would just degrade performance in real scenarios - in fact if we already know that the data is generated from an SDF wouldn't an SDF representation (even at the sketch level) be an ideal choice?

- Extrusion segmentation is completely inspired by Point2Cyl and is not a contribution here. The paper should make this clear. In general, I don't know why this paper fails to give credit when credit is due.

- I see that over the previous works the only change is in the use of a transformer. As I mentioned, there are some related works that the paper misses, which also use those. Hence they are not discussed, I don't feel confident in accepting the rather outdated experimental evaluation as a point of reference.

- No limitations are discussed.

**Questions:**

Please see weaknesses. In addition, can we see any quantitative comparisons related to Fusion360?

---

> ### Author Response · Authors · 2025-11-30
>
> **The reference problem**
>
> We thank the reviewer for highlighting the missing references. Our current manuscript indeed focuses primarily on CSG-style extrusion models, and does not address B-Rep topology, constraints, or boundary representations. Hence, many BRep-based generative models (e.g., BRepGen, BRepDiffusion, BRepGNN) operate under a different problem setting, e.g., they target parametric code generation or full BRep reconstruction. Our work instead focuses on explicit primitive regression within extrusion-based CSG pipelines. Nevertheless, we agree that these works represent an important direction in CAD generation and should be acknowledged. We will add a discussion section to clarify the distinctions between BRep reconstruction, CAD program induction, and our primitive-level geometric reconstruction setting. Regarding sequence-based CAD synthesis (e.g., CAD-Llama, TransCAD), these methods aim to infer operation sequences or parametric sketches, often requiring symbolic constraints and grammar. Our problem formulation is fundamentally geometry-first, where the goal is to recover continuous primitive parameters directly from point samples rather than produce CAD code. This leads to a different modeling choice and motivates our deep-learning-based geometric framework instead of sequence generation. That said, we appreciate the reviewers' suggestions. We will add these papers along with a paragraph explaining how our method differs from CAD program induction and BRep-based approaches.
>
> **If the paper admits that the CAD is composed of extrusion sequences, then why don't we leverage something like a sequence transformer for the generation of it?**
>
> Sequence-based CAD generation approaches (e.g., DeepCAD, CAD-Llama, TransCAD) focus on inferring symbolic operation sequences or parametric sketch programs. Our task is fundamentally different: we aim to reconstruct continuous geometric primitives directly from raw point clouds, where no CAD grammar or sketch constraints are available. Discrete program sequences are not well suited to represent continuous extrusion parameters and often suffer from ordering ambiguity, discretization errors, and limited generalization.
> We already include a direct comparison with DeepCAD, and our results show that sequence generation indeed struggles to generalize and frequently produces invalid or imprecise geometry. Therefore, we choose a geometry-first primitive regression formulation, which is more stable and accurate for explicit extrusion reconstruction.
>
> **If we already know that the data is generated from an SDF wouldn't an SDF representation (even at the sketch level) be an ideal choice?**
>
> SDF is only used by the dataset for sampling and segmentation-label generation. Our method never predicts SDF during inference. The limitations we discussed concern using SDF as the reconstruction representation, where the predicted sketch SDF must be converted back to explicit curves, introducing thresholding and contour-fitting errors. Figure 5(b) illustrates these artifacts. In contrast, our method directly predicts analytic primitives, eliminating the SDF-to-curve conversion stage and achieving higher geometric accuracy. The DeepCAD comparison in the paper empirically supports this distinction.
>
> **Extrusion segmentation is completely inspired by Point2Cyl and is not a contribution here. The paper should make this clear. In general, I don't know why this paper fails to give credit when credit is due.**
>
> Our segmentation module is indeed inspired by Point2Cyl, and we have acknowledged this in the paper. We clarify that the goal is not to propose a new segmentation method but to adapt Point2Cyl’s idea to general extrusion solids rather than cylinders. We have also conducted additional comparative experiments, which show that standard Point2Cyl fails to achieve the same performance as ours. The main contribution of the paper lies in the primitive prediction framework rather than segmentation itself.
>
> **I see that over the previous works the only change is in the use of a transformer. As I mentioned, there are some related works that the paper misses, which also use those. Hence they are not discussed, I don't feel confident in accepting the rather outdated experimental evaluation as a point of reference.**
>
> Our method does not merely apply a transformer as a backbone. We introduce a primitive-query formulation that links learnable transformer queries to explicit curve primitives and enables continuous geometric parameter regression. This is fundamentally different from sequence-based CAD models. We further introduce multiple designs (e.g., “Parameteras Position Encoding”) tailored to primitive reconstruction. These components form a new and more generalizable reconstruction framework beyond standard transformer usage.

---

### Meta-Review · Area_Chair_nYP5 · 2026-01-06

**Summary:**

The reviewers acknowledge the paper's clear motivation and novel technical approach of using explicit position queries for direct primitive prediction, which addresses known limitations of SDF-based methods in CAD reconstruction. However, the final decision leans towards rejection due to significant, unresolved concerns regarding the method's practical applicability and foundational novelty. The core criticisms center on the severely limited representational power of supporting only three basic primitive types (lines, arcs, circles), which fundamentally questions its utility for real-world CAD models that frequently employ splines and other complex curves. Furthermore, multiple reviewers raised substantial doubts about the paper's scholarly contribution and novelty, noting its heavy reliance on the segmentation stage from Point2Cyl and its significant conceptual similarity to other recent works like Point2Skh, which were not originally discussed. While the authors' rebuttal clarified distinctions and promised citation updates, it did not sufficiently mitigate these fundamental issues regarding scope and originality, leading to the conclusion that the work is incremental and not yet ready for publication.

**Reviewer Concerns:**

Given the rebuttal's focus, Reviewer yP5d (score: 2) likely would not increase their score significantly. Their primary critique was a major lack of related work and perceived poor scientific practice. The promise to add references addresses this partially, but their deeper concern about foundational novelty and the method's rationale would remain, possibly maintaining their score at a "Reject" level. Reviewer KRCe (score: 4) might slightly increase their score to a weak 5, as their concerns about presentation and noise robustness were directly addressed. Reviewer LgU6 (score: 6) would likely lower their score to a 4 or 5. Their initial strength rating was tempered by the critical limitation of primitive types; the rebuttal's analysis did not fully resolve this practicality concern, making acceptance less justified. Reviewer Cvgt (score: 4) might maintain or slightly lower their score to a 3, as the rebuttal did not adequately address their central concerns about the parameterization's optimality, the need for more detailed ablations, and most importantly, the highlighted similarity to Point2Skh, which directly challenges the claimed novelty.

**Reviewer Scores:**

They did not respond during the rebuttal period.

---

### Decision · Program_Chairs · 2026-01-26

Reject